# IGF2BP3 as a Novel Prognostic Biomarker and Therapeutic Target in Lung Adenocarcinoma

**DOI:** 10.3390/cells14151222

**Published:** 2025-08-07

**Authors:** Feiming Hu, Chenchen Hu, Yuanli He, Lin Guo, Yuanjie Sun, Chenying Han, Xiyang Zhang, Junyi Ren, Jinduo Han, Jing Wang, Junqi Zhang, Yubo Sun, Sirui Cai, Dongbo Jiang, Kun Yang, Shuya Yang

**Affiliations:** 1Department of Immunology, Air Force Medical University, No. 169, Changle West Road, Xi’an 710032, China; 17778914060@163.com (F.H.); 18579121005@163.com (C.H.); yuanli5832@163.com (Y.H.); 15389691926@163.com (L.G.); syjfly@163.com (Y.S.); hanchenying7777777@163.com (C.H.); zjq000211@163.com (J.Z.); sunyubo000103@163.com (Y.S.); caisirui@fmmu.edu.cn (S.C.); superjames1991@foxmail.com (D.J.); 2Yan’an Key Laboratory of Microbial Drug Innovation and Transformation, School of Basic Medicine, Yan’an University, Yan’an 716000, China; 3Military Medical Innovation Center, Air Force Medical University, Xi’an 710032, China; zhangxiyang199272@163.com; 4School of Basic Medicine, Air Force Medical University, No. 169, Changle West Road, Xi’an 710032, China; 15993319021@163.com (J.R.); mmmmmeidou@163.com (J.H.); wwangjing0405@163.com (J.W.)

**Keywords:** IGF2BP3, lung adenocarcinoma, therapeutic target, tumor immune microenvironment, prognostic biomarker

## Abstract

RNA-binding proteins (RBPs), particularly IGF2BP3, play critical but underexplored roles in lung adenocarcinoma (LUAD). This study investigated IGF2BP3′s clinical and functional significance using single-cell/RNA sequencing, validated by qPCR, Western blot, and immunohistochemistry. The results show IGF2BP3 was significantly upregulated in LUAD tissues and associated with advanced-stage, larger tumors, lymph node metastasis, and poor prognosis. A prognostic nomogram confirmed its independent predictive value. Functionally, IGF2BP3 knockdown suppressed proliferation, and induced G2/M arrest and apoptosis. GSEA linked high IGF2BP3 to cell cycle activation and low expression to metabolic pathways. Notably, high IGF2BP3 correlated with immune evasion markers (downregulated CD4+ effector T cells, upregulated Th2 cells), while TIDE analysis suggested a better immunotherapy response in low-expressing patients. Drug screening identified BI-2536 as a potential therapy for low-IGF2BP3 cases, supported by strong molecular docking affinity (−7.55 kcal/mol). These findings establish IGF2BP3 as a key driver of LUAD progression and a promising target for immunotherapy and precision medicine.

## 1. Introduction

Lung cancer stands as the most common form of cancer worldwide, and the leading contributor to cancer-related deaths [1]. From a histological perspective, lung cancers are broadly divided into two main categories: small-cell lung carcinoma (SCLC), which comprises 15–20% of cases, and non-small-cell lung carcinoma (NSCLC), representing a substantial 80–85% of all diagnoses [2]. Among NSCLC subtypes, lung adenocarcinoma takes the lead, making up over 40% of all lung cancer cases diagnosed globally [3]. Since its early symptoms are insidious, most patients are in advanced stages when diagnosed, leading to poor treatment and poor prognosis [4]. Although recent advances in targeted therapies and immunotherapies have significantly improved survival in some patients, the high degree of heterogeneity and drug resistance in LUAD remains a major challenge for clinical treatment [5]. Thus, a comprehensive investigation into the molecular mechanisms underlying LUAD is warranted, and the search for new biomarkers and therapeutic targets is important for improving patient prognosis.

RNA-binding proteins (RBPs) are essential molecules that interact with RNA and modulate their metabolism, stability, and translation [6]. Emerging evidence indicates that RBPs are implicated in diverse cancers, affecting the growth, death, spread, and migration of tumor cells [7]. In recent years, the involvement of RNA-binding proteins (RBPs) in lung cancer has garnered increasing interest, for example, RBPs such as RBM39, RBM4 and HNRNPA1 have been shown to promote cancer progression by regulating mRNA splicing [8,9,10]. IGF2BP3, a crucial component of the RBP family, shows elevated expression during embryonic stages, but remains minimally detectable in healthy adult tissues [11]. However, IGF2BP3 displays unusually elevated expression in several cancers, including cervical, gastric, breast, and colorectal, and is closely linked to tumor aggressiveness, metastasis, and unfavorable prognosis [12]. Notably, its expression in non-small-cell lung cancer (NSCLC) has also been described, and further functional studies have demonstrated that silencing IGF2BP3 via shRNA (shIGF2BP3) significantly reduces cell growth and colony formation in the A549 cell line, providing preliminary evidence for its role in lung cancer progression [13,14]. IGF2BP3 promotes tumor cell growth and viability by binding to the 3′UTR of target mRNAs, thereby enhancing their stability and translational efficiency [15]. In addition, IGF2BP3 promotes tumor immune escape by modulating immune cell presence in the TME [16]. Although the function of IGF2BP3 in a variety of cancers has been preliminarily revealed, its specific mechanism of action in lung adenocarcinoma and the clinical relevance of IGF2BP3 remain to be fully elucidated through additional research.

The role of IGF2BP3 in LUAD remains incompletely understood. In this research, we sought to clarify its expression patterns, and discovered that IGF2BP3 was markedly overexpressed in LUAD tissues; single-cell and RNA sequencing were employed. These results were subsequently confirmed through qRT-PCR, Western blot, and immunohistochemical techniques. Upon further investigation, it was revealed that IGF2BP3 expression levels correlate positively with patient clinical stages. Survival analysis demonstrated that IGF2BP3 serves as an independent predictor for lung adenocarcinoma prognosis, and the Normogram model constructed based on its expression level and clinical features has significant clinical value in predicting patient survival. Subsequently, our knockdown of IGF2BP3 significantly inhibited cancer cell proliferation and induced apoptosis. Gene set enrichment analysis indicated that elevated IGF2BP3 expression was significantly linked to the cell cycle pathway, while low expression was associated with the metabolic pathway, suggesting that it may affect tumor progression by regulating the cell cycle and metabolism. In addition, CD4+ effector memory T cell (Tem)-related genes were significantly down-regulated and genes associated with Th2 cells exhibited significant upregulation in the IGF2BP3 high-expression group, indicating that IGF2BP3 may be involved in tumor immune evasion by modulating immune cell activity. Further TIDE analysis suggested that patients with low IGF2BP3 expression exhibited reduced potential for immune evasion and may be more sensitive to immunotherapy. Drug sensitivity analysis (oncoppredict) showed that BI-2536 had potential therapeutic effects on patients with low IGF2BP3 expression, and AutoDock-based molecular docking analysis demonstrated a robust interaction between the compound and IGF2BP3, with a binding energy of −7.55 kcal/mol, underscoring a potent affinity. To wrap up, our research shed light on IGF2BP3’s pivotal role in the onset and progression of LUAD, while also unveiling a promising therapeutic avenue for patients battling this condition.

## 2. Materials and Methods

### 2.1. Data Download

To conduct the analysis, we retrieved the relevant clinical data and transcriptome data related to lung adenocarcinoma (LUAD) from the existing available databases. The specific operations were as follows. We sourced RNA sequencing transcriptome data for 571 lung adenocarcinoma patients from TCGA (https://portal.gdc.cancer.gov/) (accessed on 30 December 2023), encompassing 58 normal tissue samples, 513 tumor samples, and relevant clinical details. Furthermore, we analyzed lung adenocarcinoma mRNA expression profiles and clinical data sourced from the GEO database (https://www.ncbi.nlm.nih.gov/geo/) (accessed on 30 May 2024). The sources of these data cover several datasets, namely, GSE10072, GSE32863, GSE68465, GSE75037, and GSE131907 (single-cell sequencing data).

### 2.2. Differential Gene Expression Analysis

For the identification of differentially expressed genes, the “DESeq2” (Version 1.48.1) and “limma” (Version 3.64.1) R packages were employed. DEGs were |log2 FC| > 1.0 and FDR < 0.05.

### 2.3. Protein–Protein Interaction (PPI) Network and Cox Regression Analysis

PPI network analysis was carried out using the STRING database (https://cn.string-db.org/cgi/input?sessionId=bXmYsv7CnUrH&input_page_active_form=multiple_identifiers) (accessed on 3 November 2024) to elucidate the interaction relationship between the crossover genes. The visualization of the network and the identification of core genes were achieved using Cytoscape software (version 3.9.0). Additionally, predictive genes were identified through univariate Cox regression analysis using the “survival” package in R (Version 4.5.1).

### 2.4. Single-Cell Transcriptome Data Analysis

Based on the GEO dataset (GSE131907) [17], we conducted a comprehensive analysis of single-cell transcriptome data using R. Specifically, the t-SNE algorithm was employed for dimensionality reduction and to cluster and annotate all cells, and the distribution characteristics of cell subpopulations were demonstrated by visualization.

### 2.5. Heatmaps

To visualize the differences in gene expression, heatmaps of DEGs were created with the “pheatmap” (Version 1.0.13) package in R.

### 2.6. Analysis of Expression and Clinical Features of IGF2BP3

We obtained LUAD clinical data from the UCSC Xena (https://xenabrowser.net/datapages/) (accessed on 30 December 2023) and GEO databases and analyzed them using R, and suitable statistical methods in the form of tests were selected based on the data of the clinical characteristics being compared.

### 2.7. Cell Culture

The BEAS-2B human bronchial epithelial cells were grown in DMEM medium, which was enriched with 10% fetal bovine serum (FBS). On the other hand, the lung adenocarcinoma cell lines A549, H1650, and PC9 were nurtured in RPMI-1640 medium, also fortified with 10% FBS. Each cell line was verified for authenticity and regularly inspected to confirm that it was genuine and free of mycoplasma contamination.

### 2.8. Quantitative Real-Time Reverse Transcription Quantitative PCR (qRT-PCR)

To extract cellular RNA, we employed an RNA extraction kit manufactured by TSINGKE in Beijing, China. Once the RNA was successfully isolated, it was converted into cDNA through reverse transcription using the HiScript II Q RT SuperMix for qRT-PCR, a product from Vazyme Biotech in Nanjing, China, which also incorporates a gDNA elimination feature. qRT-PCR was performed using SYBR Green PCR Master Mix (Vazyme Biotech, Nanjing, China), with primer sequences provided in Appendix A.

### 2.9. Western Blot

Protein levels were assessed via Western blotting following standard procedures [16]. Once the blocking process was completed, polyvinylidene difluoride (PVDF) membranes were incubated using primary antibodies (Appendix A) at 4 °C throughout the night.

### 2.10. Nomogram Construction

To develop a predictive framework, a multivariate Cox regression analysis was utilized. The model’s performance was assessed using the ROC curve’s AUC and the C-index to measure outcome discrimination. A higher C-index indicates a more robust model with enhanced precision in predicting patient survival durations. Additionally, a nomogram was created, incorporating the key prognostic indicators, to provide estimates of survival probabilities at 1, 3, and 5 years for patients.

### 2.11. RNA Interference

We sourced IGF2BP3-specific siRNAs along with negative control (NC) siRNAs from GenePharma, based in Shanghai, China. Using Lipofectamine 2000 (Invitrogen, Carlsbad, CA, USA) as the delivery agent, we transfected the siRNAs into the cells over a 48 h incubation period. To verify the success of the transfection, we conducted qRT-PCR and Western blot assays. For precise information on the siRNA sequences, refer to Appendix A.

### 2.12. Cell Viability Assays

The Incellgene (Hangzhou, China) Cell Counting Kit-8 (CCK-8) was used to evaluate cell proliferation. Transfected cells were seeded in 96-well plates at 8000 cells per well and incubated in RPMI-1640 medium containing 10% CCK-8 reagent for 90 min. Following incubation, absorbance readings were taken at 450 nm using a microplate reader.

### 2.13. Flow Cytometry

The impact of IGF2BP3 on apoptosis and cell cycle dynamics in PC9 and H1650 cells was investigated through flow cytometry. Apoptosis was detected using an Annexin V-FITC/PI dual staining kit, while cell cycle phases were analyzed with PI/RNase staining buffer (BD, Franklin Lakes, NJ, USA).

### 2.14. Scratch Wound-Healing Assay

After transfection, the cells were seeded into a 6-well plate. Subsequently, a scratch was created using the tip of a sterile pipette. The wells were rinsed with PBS to eliminate cell debris, followed by replacement with serum-free medium. The cells’ wound healing capacity was monitored under phase-contrast microscopy at 0, 24, and 48 h following the scratch.

### 2.15. Weighted Gene Co-Expression Network Analysis

Weighted gene co-expression networks were constructed utilizing the “WGCNA” package in the R programming environment, aiming at identifying gene modules related to IGF2BP3 and their potential functions. The ideal soft thresholding parameter (β) was established using the “pickSoftThreshold” function. This step was crucial for constructing the neighbor-joining matrix and identifying the core modules. Subsequently, Pearson correlation coefficients were calculated between each module and *IGF2BP3* gene expression levels, and the most highly correlated modules were selected for in-depth analysis.

### 2.16. Immune Cell Infiltration Analysis

The infiltration of immune cells within lung adenocarcinoma tumors was assessed using the single-sample Gene Set Enrichment Analysis (ssGSEA) method, implemented through the GSVA (Version 2.2.0) R package. IGF2BP3 expression and immune cell infiltration were analyzed using gene expression data. Statistical significance was assessed via the Wilcoxon rank-sum test and Spearman’s correlation, with *p*-values determining outcomes

### 2.17. Immunotherapy and Prediction of Drug Sensitivity

Immunotherapeutic response was predicted using the TIDE algorithm, and drug sensitivity analysis was conducted with the R package “oncoPredict” (Version 1.2).

### 2.18. Molecular Docking

The 3D structure of BI-2536 was retrieved from PubChem (https://pubchem.ncbi.nlm.nih.gov/) (accessed on 13 November 2024) in the SDF format. Subsequently, it was transformed into the PDB format by means of Open Babel. The structure of the IGF2BP3 protein was obtained from the Protein Data Bank (PDB) (https://www.rcsb.org/) (accessed on 13 November 2024), and was generated into a PDB file by using AutoDock to generate PDB files after removing water molecules. Subsequently, the files were converted to PDBQT format to identify active sites. Molecular docking simulations were performed by AutoDock software (Version 1.5.6), and the results were presented visually using PyMol (version 2.6.0).

## 3. Result

### 3.1. Differential Expression Analysis of RBPs Was Performed Between LUAD and Normal Tissues

First, we performed differential expression gene analysis using TCGA-LUAD and GEO (GSE10072) datasets. Our findings revealed 5403 distinctively expressed genes in TCGA-LUAD, and corresponding volcano maps were generated (Figure 1A). Meanwhile, 1486 differential genes were identified in the GSE10072 dataset, and volcano and heat maps were generated (Figure 1B,C). Subsequently, we analyzed the intersections of these differential genes with RBPs, and finally identified 38 differentially expressed RBPs (Figure 1D). On this basis, we employed the STRING database and Cytoscape software (Version 3.9.1) (provided by the National Institute of General Medical Sciences, NIGMS, San Diego, CA, USA) to construct a PPI network. The purpose of this network was to illustrate the interaction profiles of the 29 genes (Figure 1E). In addition, we applied univariate Cox regression analysis to the 38 RBPs, and identified that 8 of them had significant relationships with the prognosis of LUAD (Figure 1F). Finally, by integrating the results of PPI network analysis and one-way COX regression analysis, we identified six RBPs that are both DEGs and associated with lung adenocarcinoma prognosis—IGF2BP3, SNRPE, GAPDH, MRPL12, MRPL15, and INTS8, with IGF2BP3 being the most differentially expressed (Figure 1G). To further explore the expression profile of IGF2BP3 in LUAD tissues, we initially utilized the single-cell dataset GSE131907 for analysis. Through this approach, we successfully resolved 13 distinct cell clusters and classified them into eight major cell types, such as B cells, T cells, and epithelial cells (Figure 1H). Next, we analyzed IGF2BP3 expression in healthy and cancerous tissues, revealing significantly higher levels in tumors compared to normal samples (Figure 1I–K). This indicates IGF2BP3 may be crucial in the initiation and progression of lung adenocarcinoma.

### 3.2. Association Between IGF2BP3 Expression and Clinical Features in LUAD

We examined the link between IGF2BP3 expression and the clinical features of LUAD patients through a thorough assessment of IGF2BP3 expression. We utilized transcriptomic data from TCGA, and complemented this with analyses of the GSE10072 dataset as well as the combined TCGA + GTEX datasets. The results indicate a notable upregulation of IGF2BP3 expression in paired lung adenocarcinoma tissues when compared to normal tissues (Figure 2A). Additionally, in LUAD tissues, IGF2BP3 levels were markedly higher in comparison to normal lung tissue (Figure 2B–D). To further validate this finding, we extracted data from two other datasets (GSE32863 and GSE75037) for analysis, and the results were consistent with the above findings (Appendix A). Significantly, IGF2BP3 levels showed a link to tumor progression stage (Figure 2E). Subsequent analysis demonstrated that IGF2BP3 levels were markedly elevated in individuals diagnosed with mid-to-late stage lung adenocarcinoma, starkly contrasting with those in the early stages of the condition. Moreover, the expression levels were higher in patients at stages T2 and T3–4 than in those at stage T1 (Figure 2F), a result that was consistent with the findings in the GSE32863 dataset (Appendix A). Moreover, advanced lymph node metastasis was linked to IGF2BP3 expression (Figure 2G). To further assess the clinical implications of IGF2BP3, we conducted a survival analysis. This study utilized the TCGA dataset and the combined TCGA + GTEx datasets, in conjunction with prognostic information. The findings indicate that LUAD patients exhibiting reduced IGF2BP3 levels had markedly extended overall survival compared to those with elevated IGF2BP3 expression (Figure 2H,I). To confirm these findings, we performed qRT-PCR and Western blot to measure IGF2BP3 mRNA and protein levels across different cell types. The data reveal a striking upregulation of IGF2BP3 in LUAD cell lines compared to normal bronchial epithelial cells (Figure 2J–L). In addition, immunohistochemical staining further revealed that IGF2BP3 staining intensity was markedly higher in LUAD tissues than in normal lung tissues, reflecting increased IGF2BP3 expression in LUAD (Figure 2M). Collectively, these results suggest that IGF2BP3 expression in LUAD is negatively correlated with patient prognosis, and may serve as a potential prognostic marker and therapeutic target.

### 3.3. Prognostic Modeling and the Clinical Value of IGF2BP3

In this study, using the “rms” package in R, we integrated data related to survival time, survival status, and six clinical characteristics. A nomogram was then constructed using Cox regression analysis to evaluate the prognostic significance of these characteristics in a cohort of 476 cases. As shown in Figure 3A, the total score was obtained by summing the patients’ scores corresponding to the six index levels in the graph, and patient survival could be evaluated using the scale provided at the bottom. The overall consistency index (C-index) of the model was 0.68 (95% CI: 0.63–0.73, *p* = 2.11 × 10^−12^), and the effect of each variable on survival was visualized by the length of the line segments. The results indicate that IGF2BP3 expression was an important factor influencing tumor-specific survival. Moreover, calibration curves were plotted to evaluate prediction model accuracy (Figure 3B). The model demonstrates promise for prognostic prediction in LUAD. The predictive performance was assessed via ROC curve analysis employing the pROC package in R, with AUC values and confidence intervals computed at 1-, 3-, and 5-year intervals (Figure 3C). The results reveal that the model’s area under the curve (AUC) metrics were 0.70, 0.71, and 0.73 at the 1-year, 3-year, and 5-year intervals, respectively, showcasing its robust predictive performance. Importantly, IGF2BP3 expression itself demonstrated significant standalone prognostic value (AUC = 0.63), rivaling key clinical parameters (T-stage = 0.65, N-stage = 0.63, Stage = 0.69), and its integration into a composite RiskScore model significantly enhanced predictive accuracy (AUC = 0.72) beyond any single variable (Appendix A). Based on the median expression level, participants were categorized into high and low IGF2BP3 expression cohorts. The data underscore that individuals with lower IGF2BP3 expression experienced markedly longer survival durations compared to their high-expression counterparts (Figure 3D). In addition, we further explored the correlation between IGF2BP3 expression and the clinical characteristics of LUAD patients. Patients were categorized into high and low IGF2BP3 expression groups based on the median expression threshold. The data show a clear and meaningful link between higher IGF2BP3 levels and more advanced tumor grades, later clinical stages, and increased lymph node metastasis in individuals with LUAD (Table 1, Figure 3E). These findings indicate that IGF2BP3 may be implicated in the progression and prognosis of LUAD.

### 3.4. Downregulation of IGF2BP3 Inhibits LUAD Cell Proliferation and Promotes Apoptosis

To investigate the role of IGF2BP3 in LUAD, we employed small interfering RNA (siRNA) to selectively knock down IGF2BP3 in PC9 and H1650 cell lines. The knockdown efficiency was confirmed by qRT-PCR (Figure 4A) and Western blot (Figure 4B,C). Functional assays demonstrated that IGF2BP3 knockdown significantly decreased LUAD cell viability, as shown by the CCK8 proliferation assay (Figure 4D). Flow cytometry analysis demonstrated that IGF2BP3 knockdown markedly enhanced the rate of apoptosis (Figure 4E,F) and triggered G2/M phase cell cycle arrest in both PC9 and H1650 cell lines (Figure 4J and Appendix A). Additionally, the expressions of apoptosis-associated genes BIM, BAX and BCL-2 were evaluated through qRT-PCR and Western blot analysis. The results show that IGF2BP3 knockdown increased BIM expression, thereby promoting apoptosis in LUAD cells (Figure 4G–I and Appendix A). Moreover, wound-healing assays showed that IGF2BP3 silencing markedly impaired LUAD cell migration (Figure 4K,L). Subsequently, we successfully reversed the upregulation of BIM mRNA caused by IGF2BP3 knockdown in PC9 and H1650 cell lines after inhibiting BIM expression by BIM-specific siRNA (Appendix A). The knockdown of BIM partially counteracted the increased apoptosis in cancer cells caused by IGF2BP3 knockdown (Appendix A). In summary, our findings indicate that IGF2BP3 knockdown suppresses LUAD cell proliferation and migration, induces G2/M phase cell cycle arrest and enhances apoptosis, thereby underscoring the potential involvement of IGF2BP3 in LUAD progression.

### 3.5. Enrichment Analysis of DEGs in High- and Low-IGF2BP3 Groups

Using the TCGA-LUAD dataset, patients were categorized into high and low IGF2BP3 expression groups based on the median expression level as the threshold, and we performed differential expression analysis. Volcano plots and heatmaps were subsequently generated (Figure 5A,B), and we initially screened out differential genes associated with IGF2BP3 expression. Subsequently, using the R package “WGCNA,” co-expression networks were built to identify gene modules related to IGF2BP3 expression. Through soft threshold screening, β = 10 was identified as the optimal parameter for constructing scale-free networks (Figure 5C,D). Following optimization of the WGCNA parameters, we employed a phylogenetic tree to delineate co-expression modules (Figure 5E), and ultimately identified eight modules, calculating the correlation between their eigengenes and IGF2BP3 expression (Figure 5F). To validate the results, we repeated the analysis using a separate dataset, GSE10072, and obtained the module with the highest correlation, containing 648 genes (Appendix A). Next, we identified differentially upregulated genes between the high and low IGF2BP3 expression groups by intersecting them with the most highly correlated genes from the WGCNA co-expression network modules. Using a Venn diagram, we identified 20 overlapping DEGs (Figure 5G). These genes may have important regulatory roles in the growth of LUAD cells. Following this, we carried out a Gene Ontology (GO) enrichment analysis on the set of 20 genes, which uncovered notable associations with key biological processes. These included the progression through the G2/M phase during the meiotic cell cycle and the transition from metaphase to anaphase in the mitotic cell cycle. The findings highlight the critical roles these genes play in cell cycle regulation, mitotic sister chromatid segregation, and nuclear chromosome segregation (Figure 5H). In addition, KEGG analysis showed the notable pathway enrichment of differentially expressed genes in the cell cycle (Figure 5I). These findings suggest that IGF2BP3 may influence LUAD progression by modulating cell cycle-related genes. To explore the functional mechanisms of IGF2BP3 in LUAD, we performed GSEA. The results demonstrate significant enrichment in cell cycle-related pathways in the IGF2BP3 high expression group, which was enriched in pathways such as HALLMARK_G2M_CHECKPOINT and HALLMARK_MYC_TARGETS_V1 (Figure 5J). Conversely, the IGF2BP3 low expression group exhibited significant enrichment in lipid metabolism-related pathways, including HALLMARK_FATTY_ACID_METABOLISM and HALLMARK_ADIPOGENESIS (Figure 5K). These results suggest that IGF2BP3 may promote tumor cell proliferation by regulating cell cycle-related genes, while its low expression may inhibit tumor progression by affecting lipid metabolic pathways. In conclusion, IGF2BP3 might be crucial in lung adenocarcinoma development through its regulation of cell cycle and metabolic pathways.

### 3.6. Association Between IGF2BP3 and Immune Cell Infiltration in LUAD

To explore the relationship between IGF2BP3 levels and the immune landscape, ssGSEA was performed on LUAD RNA-seq datasets; we then screened 32 immune cell types with abundance > 0 and assessed their infiltration levels (Figure 6A). Difference and correlation analyses revealed that two types of tumor-infiltrating immune cells (TICs) were significantly associated with IGF2BP3 expression. Specifically, Th2 cells exhibited a positive correlation with IGF2BP3 expression, whereas CD4+ effector memory T cells (Tem) showed a negative correlation with IGF2BP3 expression (Figure 6B–G and Appendix A). These findings indicate that IGF2BP3 expression levels markedly affect immune cell infiltration within the TME. Notably, Th2 cells were able to promote immune escape in the TME while CD4+ Tem exerted anti-tumor effects [18,19]. Next, we investigated IGF2BP3′s role in the tumor immune microenvironment by comparing high- and low-expression groups using GSEA. The findings reveal that genes associated with CD4+ Tem cells were significantly downregulated, while Th2 cell-related genes exhibited significant upregulation in the high IGF2BP3 expression group versus the low expression group (Figure 6H). This finding suggests that IGF2BP3 may promote tumor immune evasion by modulating the functions of these immune cells. Based on this finding, we computed the Tumor Immune Dysfunction and Exclusion (TIDE) score, which assesses the likelihood of immune evasion by analyzing the gene expression profiles of tumor samples. A reduced TIDE score suggests lower tumor immune escape risk and higher immunotherapy response potential in patients [20]. Our analysis showed that the TIDE score was significantly lower in the IGF2BP3_low compared to the IGF2BP3_high (Figure 6I). This indicates that low IGF2BP3 levels in patients could enhance immunotherapy efficacy. Collectively, these results highlight the potential role of IGF2BP3 in shaping the TIME, and provide a theoretical foundation for developing immunotherapy strategies in LUAD patients.

### 3.7. Analysis of Drug Sensitivity in LUAD Patients with High and Low IGF2BP3 Expression

To assess IGF2BP3′s role in drug therapy, we analyzed LUAD drug sensitivity using the R package “oncoppredict.” The data included 513 TCGA and 58 GSE10072 LUAD samples. The results show nine TCGA and four GSE10072 drugs were more effective in patients with reduced IGF2BP3 expression, suggesting that these drugs may have better efficacy (Figure 7A,B). Subsequently, the TCGA and GSE75037 datasets were analyzed by use of Venn diagrams for co-sensitized drugs, and one shared drug was screened, BI-2536 (Figure 7C). To further assess the binding potential of BI-2536 to IGF2BP3, we performed molecular docking analysis using AutoDock. The binding affinity reflects the ability of the ligand to bind to the receptor, and a higher absolute value indicates a stronger binding ability. The binding energy between BI-2536 and IGF2BP3 was calculated to be −7.55 kcal/mol, indicating that the combination of the two has good binding potential. The 3D model of molecular docking was drawn by PyMol (version 2.6.0) (Figure 7D). Collectively, these findings indicate that BI-2536 might offer enhanced therapeutic efficacy for LUAD patients with low IGF2BP3 expression, which provides an important basis for clinical decision-making on medication and the improvement of therapeutic precision. These findings provide new insights and a theoretical framework for creating personalized treatments for LUAD patients with low IGF2BP3 expression, especially regarding immune and pharmacological approaches.

## 4. Discussion

Lung cancer remains one of the deadliest forms of cancer globally, claiming countless lives each year and prompting extensive research into its epidemiology and treatment outcomes. Within this landscape, non-small-cell lung cancer (NSCLC) stands out, accounting for roughly 80% of all lung cancer cases. Its prevalence has made it a focal point in both medical studies and clinical practice. NSCLC mainly contains adenocarcinoma (AD), and has emerged as a significant public health challenge that poses a severe threat to human health [21,22]. Despite progress in early diagnosis and targeted therapies, LUAD’s morbidity and mortality rates persist at elevated levels, particularly for advanced-stage patients, whose prognosis remains poor [23]. It is worth noting that even with a combination of modern treatments such as conventional chemotherapeutic agents and molecularly targeted therapies, the five-year survival rate for lung adenocarcinoma patients is around 15% [24,25,26]. Thus, identifying reliable early prognostic biomarkers for personalized treatment is crucial. This highlights the need for deeper insights into LUAD’s molecular pathways and the creation of innovative treatment approaches.

RNA-binding proteins (RBPs), pivotal in post-transcriptional control, engage target RNAs via their specialized binding domains, and are implicated in RNA splicing, translocation, editing, localization, and translational regulation, thereby broadly contributing to the modulation of cellular physiological and pathological processes. Notably, numerous studies have demonstrated that RBPs regulate various tumor markers and are crucial in cancer advancement by promoting cell growth, preventing cell death, sustaining tumor stem cell traits, enhancing cell migration, and mediating immune escape [7]. As an important member of the RBP family, IGF2BP3 is intimately associated with the processes of tumorigenesis, progression and metastasis of a variety of tumors. It has been demonstrated that IGF2BP3 shows an abnormal expression pattern in a variety of malignant tumors. In particular, in gastric, esophageal squamous cell, and cervical cancers, IGF2BP3 levels are significantly elevated, showing a strong correlation with lymph node metastasis and poorer prognosis [27,28,29,30]. It has been shown that in cervical cancer, IGF2BP3 promotes lipid metabolism reprogramming by upregulating the expression of stearoyl coenzyme A desaturase (SCD), which subsequently promotes the survival and proliferation of tumor cells [29]. Furthermore, IGF2BP3 plays a part in modulating the infiltration of immune cells within the TME, thereby impacting its cellular landscape. It was found that the elevated expression of IGF2BP3 can foster the growth of hepatocellular carcinoma by regulating macrophage infiltration in the tumor microenvironment, promoting M2-type polarization, and inhibiting CD8+ T-cell activation, thus weakening the anti-tumor immune response [31]. Th2 cells and CD4+ effector memory T cells (Tem) are key immunomodulators within the TME. Th2 cells promote tumor immune escape and metastasis mainly by stimulating the activity of immunosuppressive cells [32]. Specifically, Th2 cells promote cancer cell proliferation by secreting cytokines, such as IL-4 and IL-10, driving macrophage differentiation to the M2 phenotype [33]. In contrast, CD4+ Tem cells, an important subpopulation of memory T cells, have a rapid effector function and the ability to migrate to peripheral tissues. In tumor immunity, CD4+ Tem cells directly kill cancer cells, destroy tumor vasculature, and maintain leukocyte responses by secreting cytokines such as IFNγ and TNF [34,35]. Moreover, CD4+ Tem cells stimulate the activation of natural killer (NK) cells and CD8+ T cells through the secretion of IL-21. This enhances their capacity to produce IFN-γ, a cytokine that is indispensable for the antitumor function of TH9 cells [36]. Building on the aforementioned research context, our analysis unveiled a positive association between IGF2BP3 and Th2 cells; contrastingly, a negative correlation was discerned with CD4+ Tem cells. Further GSEA analysis indicated that genes associated with CD4+ Tem cells were significantly downregulated, whereas elevated IGF2BP3 expression correlated with increased Th2-related gene activity, suggesting its potential role in lung adenocarcinoma by influencing immune cell infiltration. Polo-like kinase 1 (PLK1), a highly conserved serine/threonine kinase, plays a pivotal role in numerous mitotic processes, encompassing mitotic entry, spindle assembly, and the segregation of chromosomes [37,38]. Given its critical role in cell cycle regulation, the inhibition of PLK1 activity has been shown to significantly inhibit the proliferation of a wide range of cancer cell lines, thus emphasizing its considerable promise as a therapeutic target for cancer therapy. For example, by targeting PLK1, the small molecule inhibitor BI-2536 not only inhibited cancer cell proliferation, but also induced mitochondrial fusion, G2/M-phase blockade, and apoptosis in neuroblastoma cells [39,40]. On this basis, our study further explored the potential mechanism of action of PLK1 inhibitors. Through molecular docking, we found that BI-2536 was able to bind to IGF2BP3, which may provide a new molecular basis for its anti-tumor function. In addition, we found that siIGF2BP3 inhibited LUAD cell proliferation. It also caused G2/M blockade, induced apoptosis in LUAD cells, and promoted apoptosis in LUAD cells, a result that further supports the hypothesis that BI-2563 may exert its anti-tumor effects by targeting IGF2BP3. Taken together, these findings not only reveal a potential new mechanism of BI-2563 in tumor therapy, but also provide new ideas and directions for the future precision therapy of lung adenocarcinoma.

Although our results confirm that IGF2BP3 is a key driver of lung adenocarcinoma progression and a potential therapeutic target, its role needs to be elucidated in the context of other genes that have been clearly identified as having prognostic or therapeutic significance in LUAD. The most widely studied prognostic markers in LUAD include genes involved in growth factor signaling and oncogenic pathways. For example, EGFR mutations are well-recognized drivers associated with advanced disease and poor prognosis, and are key targets for tyrosine kinase inhibitors (TKIs) such as gefitinib [41]. Similarly, KRAS mutations are associated with tumorigenesis and prognosis [42]. Another class of related genes with similar roles is RNA-binding proteins (RBPs), such as the widely expressed HuR (ELAVL1), whose expression is upregulated in tumors and promotes tumor growth [43]. Although ELAVL1 and IGF2BP3 are both RBPs and have pro-tumorigenic functions, the specific association of IGF2BP3 with G2/M-phase cell cycle arrest in regulating immune cell infiltration (e.g., down-regulation of CD4+ effector memory T cells, up-regulation of Th2 cells) as well as in functional assays sets it apart from these RBPs, highlighting the unique intersection between post-transcriptional regulation and immune escape, which undergo a unique cross-talk. In the context of immunotherapy and personalized medicine, genes like PD-L1 have gained prominence. PD-L1 expression predicts responses to immune checkpoint inhibitors (ICIs) in LUAD, with high expression indicating better ICI efficacy [44]. STK11 mutations, conversely, are linked to immune-cold tumors and resistance to ICIs [45]. Our TIDE analysis showed that a low expression of IGF2BP3 was associated with good immunotherapy response; this is similar to the predictive value of PD-L1, which regulates the tumor immune microenvironment through Th2/CD4+ effector memory T-cell imbalance, while PD-L1 suppresses the immune response through direct interaction with T-cell receptors. This suggests that IGF2BP3 may serve as a complementary biomarker for PD-L1. The above features suggest that IGF2BP3 provides new insights for personalized therapy and prognostic assessment.

This study emphasizes the crucial role and clinical relevance of IGF2BP3 in LUAD. IGF2BP3 was identified as significantly upregulated in LUAD tissues, and exhibited a positive correlation with tumor stage, T stage, and lymph node metastasis, highlighting its involvement in LUAD progression. Survival analysis revealed IGF2BP3 as an independent prognostic marker. The Nomogram prognostic model, built based on its expression and clinical features, demonstrated robust predictive performance, offering a valuable tool for clinical prognostic evaluation. Next, we knocked down IGF2BP3, which significantly inhibited lung adenocarcinoma cell proliferation and induced apoptosis, suggesting its pro-carcinogenic function. GSEA analysis indicated that high IGF2BP3 expression correlated with cell cycle pathway enrichment, whereas low expression was linked to metabolic pathways, suggesting that it may influence tumor progression by regulating cell cycle and metabolic reprogramming. Furthermore, the high expression of IGF2BP3 was linked to alterations in the TIME, characterized by the downregulation of CD4+ effector memory T-cell (Tem)-related genes and the upregulation of Th2-cell-related genes. This immune modulation may represent a pivotal mechanism through which IGF2BP3 facilitates immune escape. Based on these findings, TIDE analysis suggests that patients exhibiting low IGF2BP3 expression may derive greater benefit from immunotherapy. By “oncoppredict”, BI-2536 has potential therapeutic value for patients with low IGF2BP3 expression, which provides a reference for precise clinical treatment—not only as a prognostic biomarker and immunotherapy predictor, but also as a potential therapeutic target in lung adenocarcinoma treatment. These findings offer a novel foundation for personalized LUAD treatment strategies, necessitating further exploration of the molecular mechanisms by which IGF2BP3 regulates the TIME, as well as its clinical translational potential.

## 5. Conclusions

In summary, our study underscores the critical role of IGF2BP3 in LUAD progression, highlighting its potential as a prognostic marker and therapeutic target. Experimental data demonstrate that IGF2BP3 knockdown significantly inhibits the proliferation of lung adenocarcinoma cells, enhances apoptosis, and induces G2/M phase cell cycle arrest. Additionally, high IGF2BP3 expression was linked to advanced tumor stage, increased cell proliferation, and immune escape through modulation of the tumor immune microenvironment. Conversely, low IGF2BP3 expression predicted better immunotherapy response and potential sensitivity to BI-2536. Therefore, IGF2BP3 could be a promising target for personalized therapy, offering a new strategy for precision medicine in LUAD treatment.

## Figures and Tables

**Figure 1 cells-14-01222-f001:**
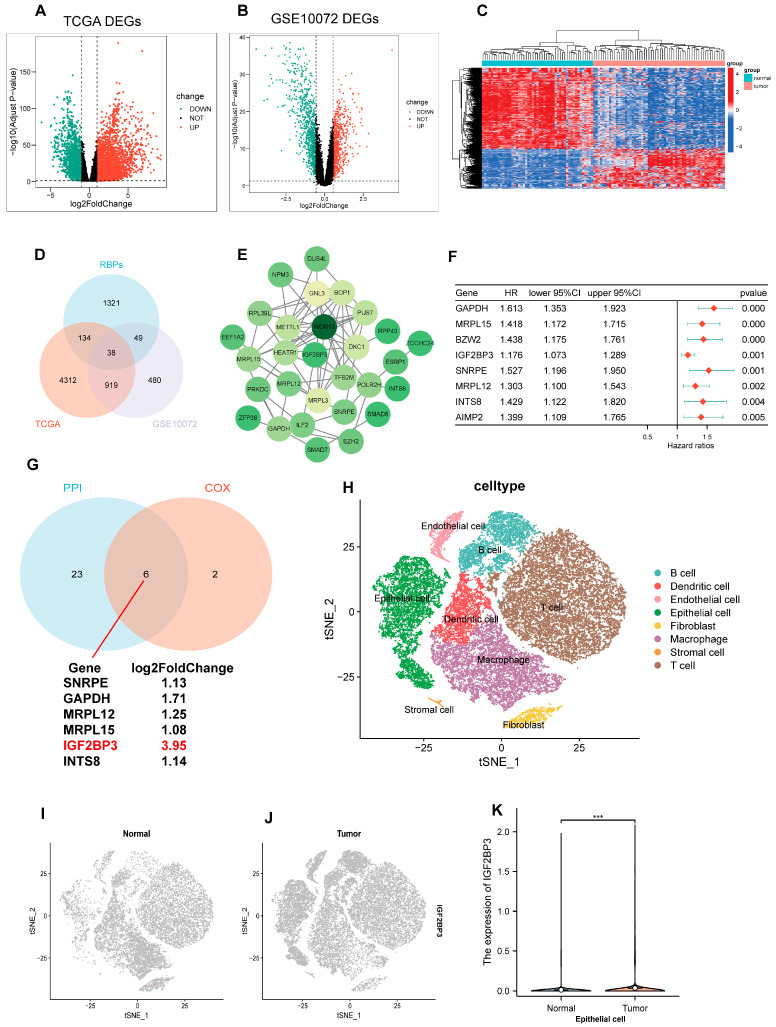
Differential expression analysis of RBPs was performed between LUAD and normal tissues. (**A**–**C**) Volcano and heatmap plots illustrate differentially expressed genes (DEGs) characterized. (**D**) Venn diagram illustrates the overlapping genes among RBPs, TCGA, and GSE10072 datasets. (**E**) The STRING database was used to create the PPI network. (**F**) Univariate Cox regression identified prognostic genes within the overlapping gene set. (**G**) The Venn diagram illustrates overlapping genes in PPI network nodes and univariate Cox regression analysis. (**H**) The t-SNE plot illustrates the distribution and annotation of distinct clusters. (**I**–**K**) Expression and distribution of IGF2BP3 in normal epithelial cells and tumor epithelial cells. *** *p* < 0.001.

**Figure 2 cells-14-01222-f002:**
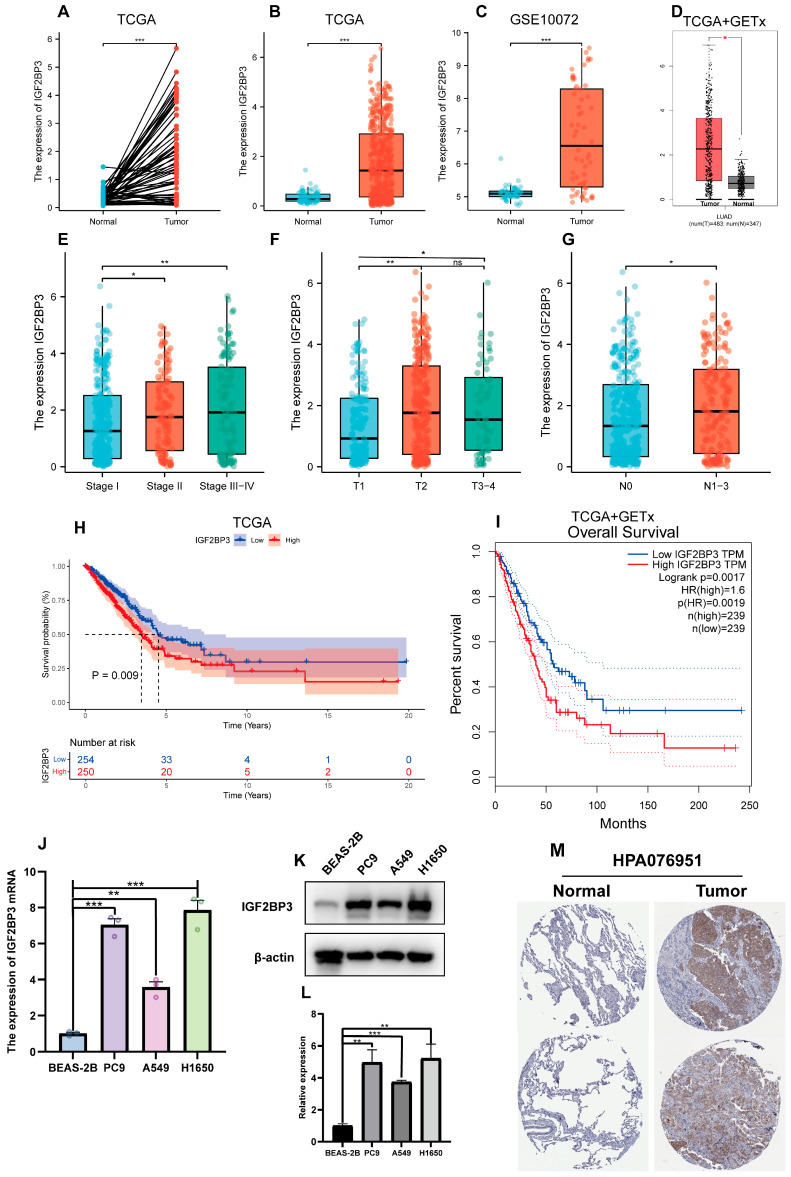
Association between IGF2BP3 expression and clinical features in LUAD. (**A**) IGF2BP3 levels were examined in paired normal and cancerous tissues from individual patients. (**B**–**D**) IGF2BP3 expression across all normal and tumor samples available. (**E**–**G**) IGF2BP3 expression in samples categorized according to tumor stage, T stage, lymph node metastasis status. (**H**,**I**) Kaplan–Meier curves were generated to assess survival differences between high- and low-IGF2BP3 expression groups, categorized by median levels. (**J**) IGF2BP3 mRNA levels in bronchial epithelium compared to LUAD lines. (**K**,**L**) IGF2BP3 protein levels were analyzed in bronchial epithelial cells and lung adenocarcinoma cell lines. (**M**) The expression of IGF2BP3 protein in LUAD tissues and normal lung tissues was evaluated by utilizing data from the Human Protein Atlas (HPA) database. (https://www.proteinatlas.org/, accessed on 30 May 2024). * *p* < 0.05, ** *p* < 0.01, *** *p* < 0.001; ns: not significant.

**Figure 3 cells-14-01222-f003:**
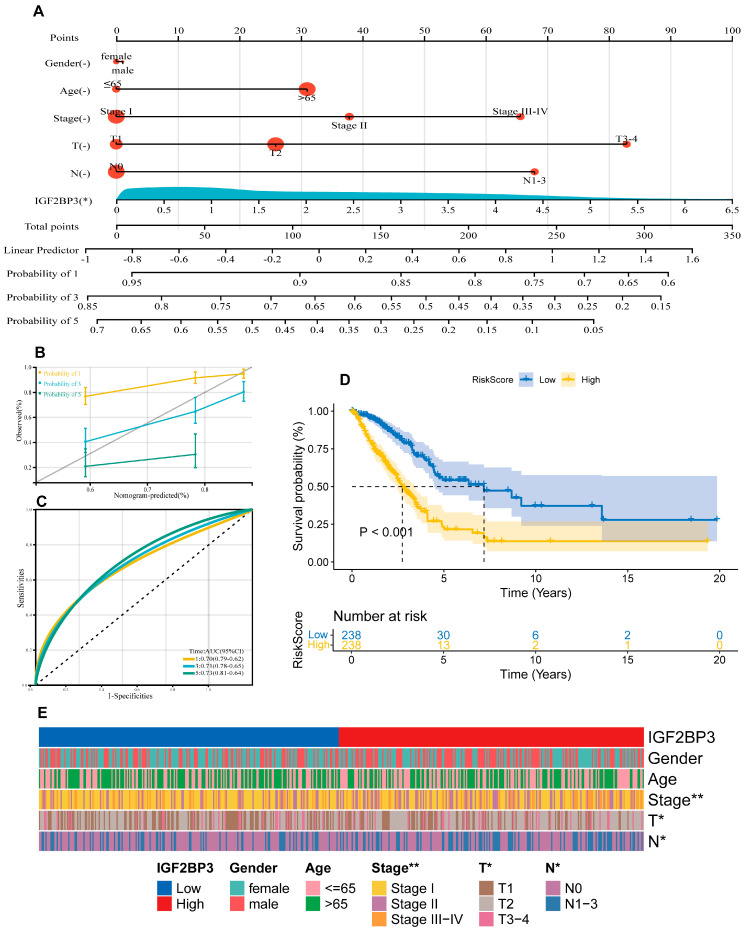
Prognostic modeling and the clinical value of IGF2BP3. (**A**) Development of a prognostic nomogram incorporating IGF2BP3 expression and multiple clinical variables. (**B**) Calibration curves assessing the prognostic model’s precision in estimating survival outcomes. (**C**) ROC curves depict the AUC for predicting overall survival rates at 1, 3, and 5 years for LUAD patients. (**D**) Kaplan–Meier analysis assessed survival differences between high-risk and low-risk patient groups, categorized based on the median RiskScore, with log-rank test indicating significant differences (*p* < 0.001). (**E**) Heatmap of clinical features. * *p* < 0.01, ** *p* < 0.01.

**Figure 4 cells-14-01222-f004:**
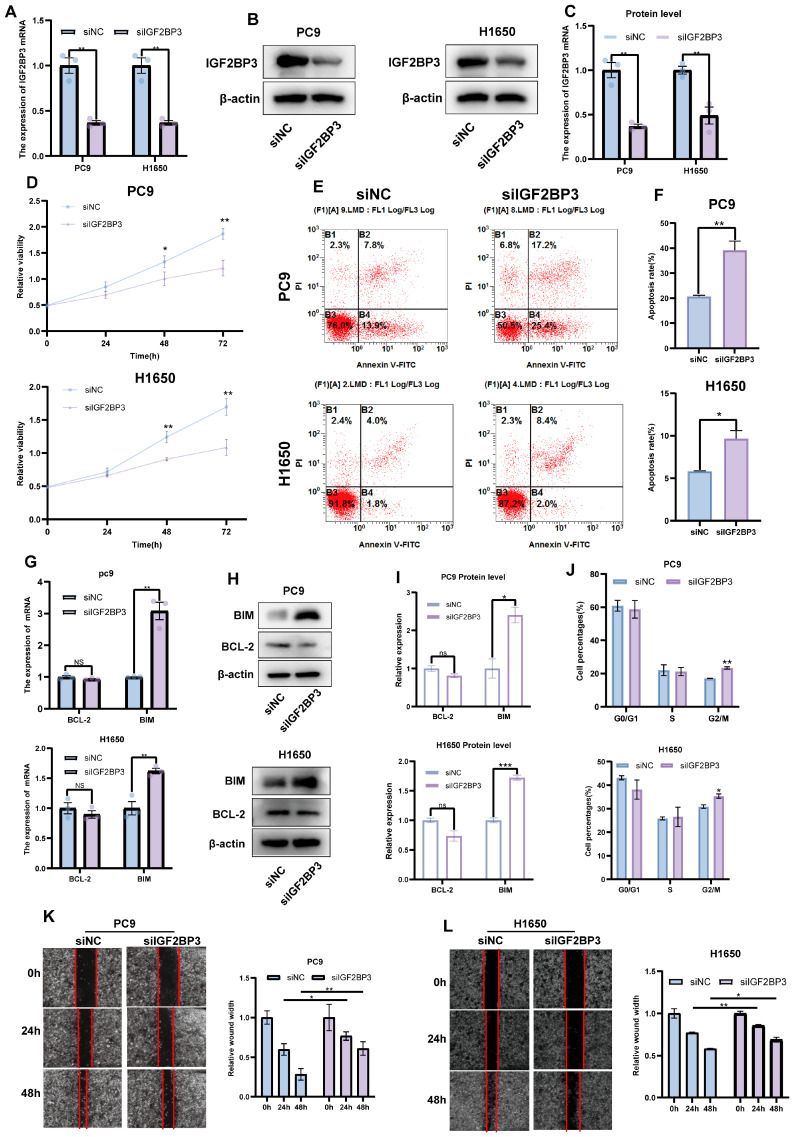
The downregulation of IGF2BP3 inhibits LUAD cell proliferation and promotes apoptosis. (**A**–**C**) IGF2BP3 siRNA transfection in PC9 and H1650 cells was followed by the assessment of IGF2BP3 mRNA and protein expression via qRT-PCR and Western blot. (**D**) Cell viability was evaluated following IGF2BP3 knockdown. (**E**,**F**) PC9 and H1650 cells transfected with IGF2BP3 siRNA were subjected to flow cytometry analysis to measure apoptosis. (**G**–**I**) Apoptosis-related genes’ *BIM* and *BCL-2* expression levels were analyzed via qRT-PCR and Western blot following IGF2BP3 suppression. (**J**) PC9 and H1650 cells transfected with IGF2BP3 siRNA were analyzed by flow cytometry to assess cell cycle distribution. (**K**,**L**) Cell migration was assessed using wound-healing assays. * *p* < 0.05, ** *p* < 0.01, *** *p* < 0.001; ns, not significant.

**Figure 5 cells-14-01222-f005:**
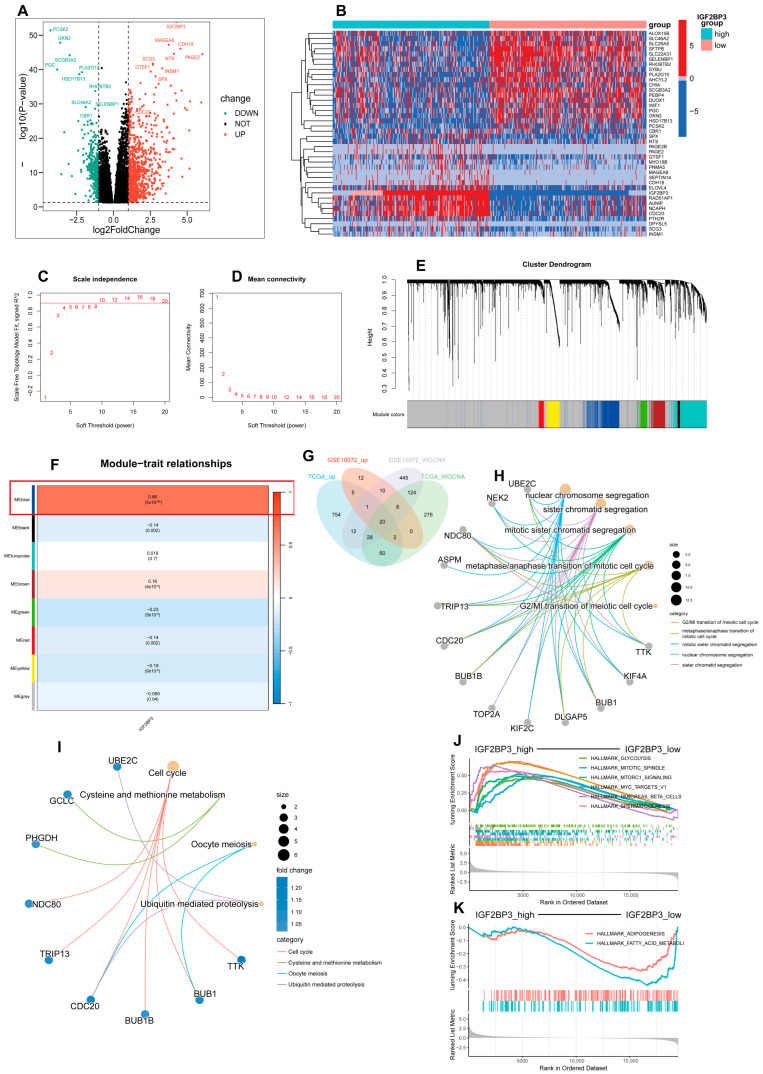
Enrichment analysis of DEGs in high- and low-IGF2BP3 groups. (**A**) Volcano plot illustrating differentially expressed genes in LUAD based on IGF2BP3 expression levels. (**B**) A heatmap illustrates differentially expressed genes (DEGs) in groups with high versus low IGF2BP3 expression. (**C**) The scale-free topology fit index was evaluated across multiple soft threshold powers (β). (**D**) Evaluation of mean connectivity across various soft threshold powers (β). (**E**) Hierarchical clustering dendrogram showing gene expression modules, with each color representing a distinct gene module. (**F**) The relationship between module eigengenes and IGF2BP3 expression was evaluated using Pearson correlation. (**G**) Venn diagram depicting the intersection of upregulated genes from TCGA, GSE10072, and DEGs linked to WGCNA. (**H**) Circos plot depicting Gene Ontology (GO) enrichment analysis of 20 DEGs, highlighting significantly enriched terms. (**I**) Circos plot of KEGG pathway enrichment analysis for 20 DEGs, with significantly enriched terms. (**J**) Gene set enrichment analysis (GSEA) showing enrichment of high-IGF2BP3-expressing samples in HALLMARK gene sets. (**K**) GSEA showing enrichment of low-IGF2BP3-expressing samples in HALLMARK gene sets.

**Figure 6 cells-14-01222-f006:**
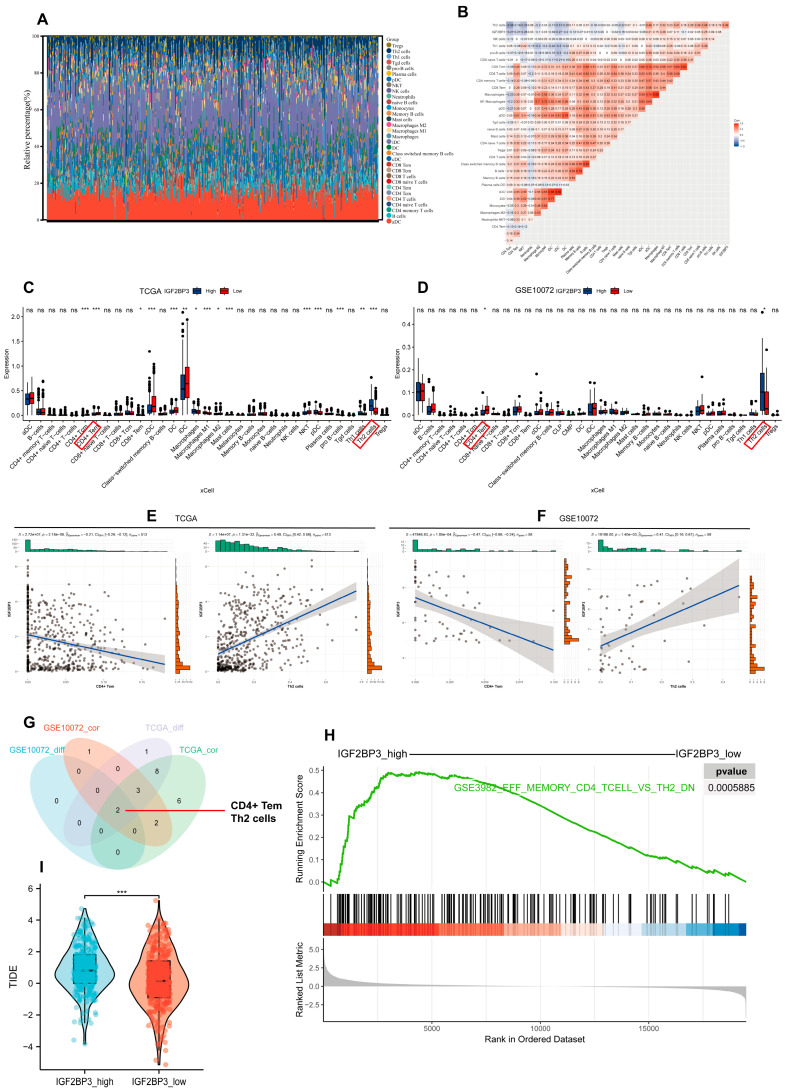
Association between IGF2BP3 and immune cell infiltration in LUAD. (**A**) Stacked bar graphs illustrating the proportions of 32 tumor-infiltrating immune cells in LUAD tumor samples. (**B**) Heatmap depicting the correlation between tumor-infiltrating lymphocytes and IGF2BP3 expression, where each color square represents the corresponding correlation value. (**C**,**D**) Box plots illustrate tumor-infiltrating immune cell distribution in LUAD samples, stratified by IGF2BP3 expression levels. The red box indicates the immune cell types that appear in both the TCGA and GSE10072 datasets and show significant differences. (**E**,**F**) Correlation analysis between IGF2BP3 expression and Th2 cells as well as CD4+ Tem cells. (**G**) Venn diagram displaying immune infiltration cells that exhibit differential expression and are associated with IGF2BP3. (**H**) GSEA results derived from the MSigDB database C7 collection. (**I**) TIDE scores for the high IGF2BP3 expression group. * *p* < 0.05, ** *p* < 0.01, *** *p* < 0.001; ns: not significant.

**Figure 7 cells-14-01222-f007:**
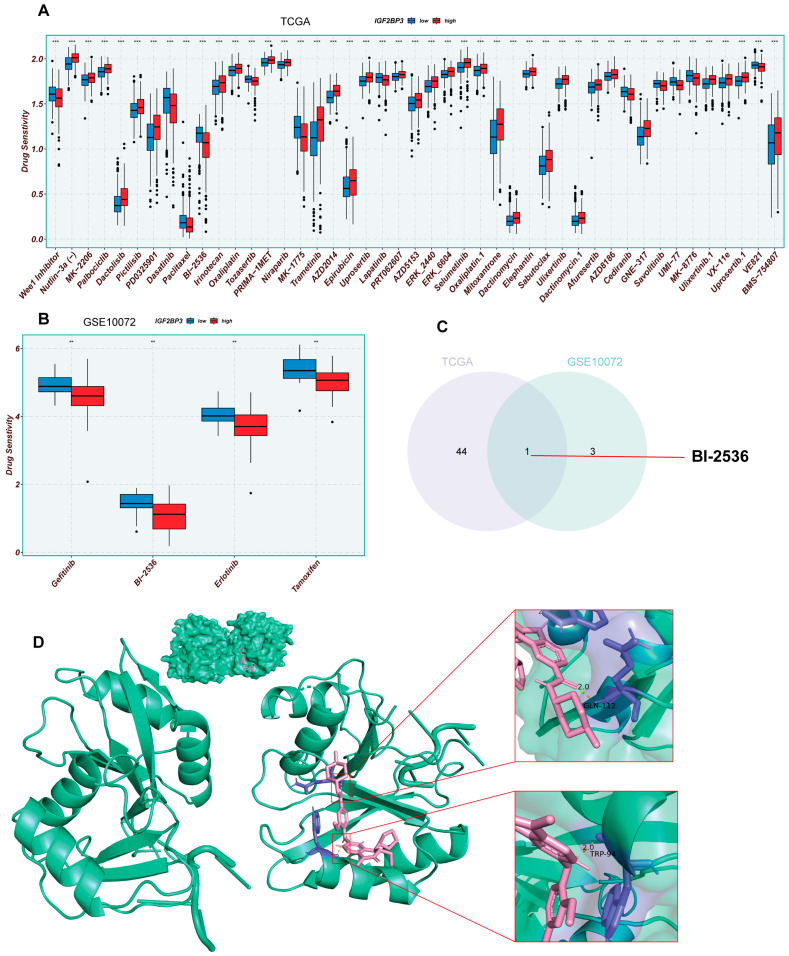
Analysis of drug sensitivity in LUAD patients with high and low IGF2BP3 expression. (**A**,**B**) Analysis of drug responsiveness in lung adenocarcinoma cases using TCGA and GSE75037 data. (**C**) Venn diagram displaying the overlap of drugs sensitive to IGF2BP3 between TCGA and GSE75037 datasets. (**D**) Molecular docking model of IGF2BP3 and BI-2536. ** *p* < 0.01, *** *p* < 0.001.

**Table 1 cells-14-01222-t001:** Comparison of clinical parameters between IGF2BP3 high and low expression groups in Lung Adenocarcinoma.

Clinical Parameters	IGF2BP3_High (N = 206, %)	IGF2BP3_Low (N = 269, %)	*p*-Value
Gender			
Male	98 (20.63%)	122 (25.68%)	0.70
Female	108 (22.74%)	147 (30.95%)	
Age			
≤65	104 (21.89%)	124 (26.11%)	0.39
>65	102 (21.47%)	145 (30.53%)	
Stage			
I	99 (20.84%)	162 (34.11%)	0.02
II	53 (11.16%)	60 (12.63%)	
III–IV	54 (11.37%)	47 (9.89%)	
T classification			
T1	53 (11.16%)	106 (22.32%)	0.007
T2	126 (26.53%)	132 (27.79%)	
T3–4	27 (5.68%)	31 (6.53%)	
N classification			
N0	124 (26.11%)	189 (39.79%)	0.03
N1–3	82 (17.26%)	80 (16.84%)	

## Data Availability

The data are available from the corresponding author upon reasonable request.

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
