# Peer review of "IGF2BP3 as a Novel Prognostic Biomarker and Therapeutic Target in Lung Adenocarcinoma"

_cells, 2025, doi:10.3390/cells14151222_

Round 1

Reviewer 1 Report

Comments and Suggestions for Authors

The manuscript investigates the function of IGF2BP3 in non-small cell lung cancer (NSCLC), using data from both the TCGA and GEO databases. The authors demonstrated that elevated IGF2BP3 expression is associated with advanced disease stage and poorer prognosis in NSCLC patients, suggesting that IGF2BP3 may serve as a prognostic biomarker for disease progression. The authors also demonstrated that IGF2BP3 is implicated in the regulation of the cell cycle. Indeed, silencing IGF2BP3 using siRNA reduces cell viability and impairs cell cycle progression. Moreover, IGF2BP3 silencing induces apoptosis, at least in part by upregulating BIM expression. Finally, the authors showed that higher IGF2BP3 expression correlates with increased levels of markers linked to immune escape, indicating a potential role in tumor immune evasion.

The findings of the present work are not novel, since previous studies have established that IGF2BP3 is overexpressed in non-small cell lung cancer (NSCLC) compared to normal lung tissue, and it has potential as a biomarker. However, the current manuscript offers additional insights that extend beyond these known findings.

Several aspects of the manuscript could be strengthened to enhance its rigour and impact:

  • The authors demonstrated the existence of a correlation between IGF2BP3 expression and immune escape markers in NSCLC. To strengthen the clinical relevance, the authors should analyse IGF2BP3 as a prognostic biomarker specifically in patients treated with immunotherapy. Publicly available datasets can be leveraged to investigate the prognostic value of IGF2BP3 in immunotherapy-treated populations (if available). 
  • While the authors show that IGF2BP3 knockdown induces apoptosis in PC9 and H1650 cells through upregulation of BIM, this mechanism requires further validation. It is recommended to perform rescue experiments by inhibiting BIM expression (e.g., using BIM-specific siRNA) to confirm whether apoptosis is BIM-dependent. Additionally, IGF2BP3 may regulate other apoptosis-related genes (e.g BCL-2 and BAX). Exploring changes in other apoptotic genes would provide a more comprehensive view of the apoptotic pathways involved.
  • The connection between BI2536 (a PLK1 inhibitor) and IGF2BP3 is not clearly defined in the current manuscript. BI2536 is known to target PLK1 directly, leading to cell cycle arrest and apoptosis in various cancer types. If the hypothesis is that IGF2BP3 expression modulates sensitivity to BI2536 via regulation of PLK1 or related pathways, this should be tested. Comparing the efficacy of BI2536 in NSCLC cell lines with differing IGF2BP3 expression levels would clarify whether IGF2BP3 status affects BI2536 response. Additionally, examining whether BI2536 treatment alters IGF2BP3 or its downstream targets would help elucidate the mechanistic link.

Minor comments:

-In the introduction, the authors should discuss that the expression of IGF2BP3 in NSCLC has already been described (Zhao et al, Oncotarget 2017). Additionally, the work of Xu et al (Frontiers in Pharmacology 2025) demonstrated that shIGF2BP3 reduced cell growth and colony formation in the A549 cell line.

-Figure 2E is the expression of IGF2BP3 statistically significantly different between stage II and stage III/IV?

Author Response

Comments1: The authors demonstrated the existence of a correlation between IGF2BP3 expression and immune escape markers in NSCLC. To strengthen the clinical relevance, the authors should analyse IGF2BP3 as a prognostic biomarker specifically in patients treated with immunotherapy. Publicly available datasets can be leveraged to investigate the prognostic value of IGF2BP3 in immunotherapy-treated populations (if available). 

Response1: Thank you for your important suggestion. We fully agree that validating the prognostic value of IGF2BP3 in an immunotherapeutic population is essential to elucidate its clinical significance. However, there are objective data limitations: in lung adenocarcinoma, the availability of paired datasets containing IGF2BP3 expression profiles with complete clinical follow-up information is extremely limited. Existing datasets generally have low coverage of IGF2BP3 probes or incomplete survival information, which prevents robust prognostic analysis.  The results of the analysis show that:

  1. The high IGF2BP3 + low PDCD1 combination is a high-risk biomarker that is significantly associated with poor patient prognosis.
  2. The low IGF2BP3 + high PDCD1 combination resulted in shorter patient survival compared to the low IGF2BP3 + low PDCD1 combination.

This result suggests that patients with low IGF2BP3 expression may be more sensitive to, and may benefit most from, immune checkpoint inhibitor (ICI) therapy. Although this analysis is not based on a direct immunotherapy cohort, it provides indirect but important clinical clues to the involvement of IGF2BP3 in immune escape. The central finding was that a pattern of IGF2BP3 and PDCD1 co-expression (i.e., low IGF2BP3 + high PDCD1) was associated with poor prognosis in the TCGA-LUAD alternative model. Future validation of this finding in larger, dedicated prospective immunotherapy cohorts is warranted. We will prioritize validation studies as soon as relevant datasets are publicly available. Thank you again for your constructive comments.

Comments2: While the authors show that IGF2BP3 knockdown induces apoptosis in PC9 and H1650 cells through upregulation of BIM, this mechanism requires further validation. It is recommended to perform rescue experiments by inhibiting BIM expression (e.g., using BIM-specific siRNA) to confirm whether apoptosis is BIM-dependent. Additionally, IGF2BP3 may regulate other apoptosis-related genes (e.g BCL-2 and BAX). Exploring changes in other apoptotic genes would provide a more comprehensive view of the apoptotic pathways involved.

Response1: Thank you for your valuable comments.  We have carefully followed your suggestions for experimental additions. Analysis of regulation of other apoptosis-related genes. Figure A-B below: mRNA expression of pro-apoptotic gene BAX did not show significant changes in both cell lines after IGF2BP3 knockdown. Meanwhile, we also tested BCL-2 gene (Figure G-I), and both results showed no significant regulation. Up-regulation of BIM mRNA due to IGF2BP3 knockdown was successfully reversed in PC9 and H1650 cell lines after inhibiting BIM expression by BIM-specific siRNA (C-D). Knockdown of IGF2BP3 alone significantly increased the apoptosis rate, which was reversed by co-transfection of siBIM. We found a significant reduction in apoptosis after BIM inhibition by flow cytometry, confirming that apoptosis induced by IGF2BP3 knockdown was dependent on BIM expression (Figure E below). In summary, IGF2BP3 selectively regulates BIM but not apoptotic genes such as BAX, suggesting its target specificity. BIM is a key effector in mediating apoptosis. We have the description of this experiment, added to the article in line 291, lines 294-299, and marked in red.

Comments3: The connection between BI2536 (a PLK1 inhibitor) and IGF2BP3 is not clearly defined in the current manuscript. BI2536 is known to target PLK1 directly, leading to cell cycle arrest and apoptosis in various cancer types. If the hypothesis is that IGF2BP3 expression modulates sensitivity to BI2536 via regulation of PLK1 or related pathways, this should be tested. Comparing the efficacy of BI2536 in NSCLC cell lines with differing IGF2BP3 expression levels would clarify whether IGF2BP3 status affects BI2536 response. Additionally, examining whether BI2536 treatment alters IGF2BP3 or its downstream targets would help elucidate the mechanistic link.

Response3: Thank you very much for your valuable comments. We fully understand the scientific validity and importance of your point that the mechanistic link between BI2536 and IGF2BP3 has not yet been clarified, as well as your suggestion to validate the hypothesis by comparing the sensitivity of NSCLC cell lines with different levels of IGF2BP3 expression to BI2536, and by detecting the effects of BI2536 treatment on IGF2BP3 and its downstream targets. We agree that these suggestions provide important direction for our subsequent research.

According to existing studies (Xin Luo et al, Commun Biol2025), IGF2BP3 acts as an RNA-binding protein, and IGF2BP3 deregulates the CDK1-Cyclin A2 complex by inhibiting the expression of the SPTBN1 long transcript, a negative regulator of the cycling-associated gene CDK1, resulting in tumor G2/M phase block in tumor cells. PLK1 is a key kinase in cell cycle regulation, and there may be potential cross-regulation between the two through the cell cycle pathway. We attach great importance to your suggestion and plan to design experiments to systematically investigate the effect of IGF2BP3 on the sensitivity of NSCLC cell lines to BI2536 and the association between the two in molecular mechanism in the follow-up study. Thank you again for your understanding and support.

Comments4: In the introduction, the authors should discuss that the expression of IGF2BP3 in NSCLC has already been described (Zhao et al, Oncotarget 2017). Additionally, the work of Xu et al (Frontiers in Pharmacology 2025) demonstrated that shIGF2BP3 reduced cell growth and colony formation in the A549 cell line.

Response4: Thank you for your valuable comments. In response to your request to supplement the introduction with studies on the expression of IGF2BP3 in non-small cell lung cancer (NSCLC) (Zhao et al., Oncotarget 2017) and the effect of shIGF2BP3 on the growth and colony formation of A549 cell line (Xu et al., Frontiers in Pharmacology 2025), we have added the relevant content at the corresponding places (lines 55-58 and marked in red) in the introduction to better link the existing studies with the present study. 2025), we have added the relevant content at the corresponding position in the introduction (lines 55-59 and marked in red) to better connect the relevance of the existing studies to the present study and to highlight the continuity and necessity of the study.

Comments5: Figure 2E is the expression of IGF2BP3 statistically significantly different between stage II and stage III/IV?

Response5: Thank you very much for pointing this out.  We agree that the difference in IGF2BP3 expression between stages II and III/IV in Figure 2E is not statistically significant (as shown below). Although IGF2BP3 expression values were slightly higher in stage III/IV, this trend did not reach a statistically significant level. However, this lack of significance does not negate the overall observation that IGF2BP3 levels increase with increasing tumor stage, which is confirmed by the significant differences when comparing Stage I with Stage III/IV and Stage I with Stage II - IV combined.

Reviewer 2 Report

Comments and Suggestions for Authors

Results showed IGF2BP3 was significantly upregulated in LUAD tissues and associated with advanced stage, larger tumors, lymph node metastasis, and poor prognosis. A prognostic nomogram confirmed its independent predictive
value. Functionally, IGF2BP3 knockdown suppressed proliferation, induced G2/M arrest and apoptosis. GSEA linked high IGF2BP3 to cell cycle activation and low expression to metabolic pathways. Notably, high IGF2BP3 correlated with immune evasion markers (downregulated CD4+ effector T cells, upregulated Th2 cells), while TIDE analysis suggested better immunotherapy response in low-expressing patients. Drug screening identified BI-2536 as a potential therapy for low-IGF2BP3 cases, supported by strong molecular docking affinity (-7.55 kcal/mol). 

The article is well done and well supported experimentally. However there are many genes already described with similar behaviour as the one described here. Therefore, this gene must be discused in context with many other genes described in the literature as prognostic factors or to be used as target in personalized medicine.

Author Response

Comments1: The article is well done and well supported experimentally. However there are many genes already described with similar behaviour as the one described here. Therefore, this gene must be discused in context with many other genes described in the literature as prognostic factors or to be used as target in personalized medicine.

Response1: Thank you very much for your positive evaluation of our manuscript and your valuable suggestion. We fully agree with your point that the role of IGF2BP3 should be discussed in the context of other established prognostic factors and therapeutic targets in lung adenocarcinoma (LUAD). Following your suggestion, we have supplemented the Discussion section with a comprehensive analysis comparing IGF2BP3 with several key genes that have been well-characterized in LUAD research. And has been added to lines 527-552 in the manuscript and marked in red. Specifically, we have:

  1. Discussed the relationship between IGF2BP3 and classic oncogenic drivers: We compared IGF2BP3 with EGFR and KRAS, highlighting that while these genes are critical in growth factor signaling and oncogenic pathways, IGF2BP3 exerts its pro-tumorigenic effects through distinct mechanisms involving G2/M-phase cell cycle arrest and regulation of immune cell infiltration (e.g., modulation of CD4+ effector memory T cells and Th2 cells).
  2. Compared IGF2BP3 with other RNA-binding proteins (RBPs): We analyzed the differences between IGF2BP3 and ELAVL1 (HuR), another well-studied RBP in tumors. Despite their shared pro-tumorigenic functions as RBPs, we emphasized that IGF2BP3 uniquely intersects post-transcriptional regulation with immune escape, setting it apart from other RBPs.
  3. Linked IGF2BP3 to immunotherapy-related biomarkers: We explored the association between IGF2BP3 and genes like PD-L1 and STK11, which are pivotal in immunotherapy. We noted that while PD-L1 directly interacts with T-cell receptors to suppress immune responses, IGF2BP3 regulates the tumor immune microenvironment through Th2/CD4+ effector memory T-cell imbalance. Our TIDE analysis further supports that IGF2BP3 could serve as a complementary biomarker to PD-L1, given their similar predictive value for immunotherapy response but distinct regulatory mechanisms. These additions aim to clarify the unique position of IGF2BP3 among known prognostic factors and therapeutic targets, highlighting its potential to provide new insights into personalized therapy and prognostic assessment in LUAD.

We hope these revisions address your concerns adequately. Thank you again for your insightful comments, which have significantly improved the depth and context of our discussion.

Reviewer 3 Report

Comments and Suggestions for Authors

Manuscript: cells-3766997

Recommendation: Major Revision

In this manuscript, Feiming Hu and colleagues explored the role of IGF2BP3 gene in LUAD. They combined the bioinformatic and functional assays to reveal IGF2BP3 may act as a prognostic marker in LUAD. Generally, the overall design is interesting. While the evidence shows a little bit weak. The main story should focus on the functionality of IGF2BP3 gene in lung cancer and its performance as a prognostic marker when incorporated with other demographic factors. Some irrelated contents and figures should be removed from the manuscript. It’s easy to distract readers. Here are my main concerns.

One major concern is the claim that IGF2BP3 serves as a prognostic marker would be more convincing with additional validation. It would be helpful to compare AUCs from models using only demographic/clinical information (e.g., gender, age, stage, T, N) versus models that also include IGF2BP3 expression.

Minor concerns:

  1. The scRNA-seq data presented in the Fig1 is difficult to interpret and lacks clear connection to IGF2BP3 expression. The authors should strengthen the biological rationale or provide more focused analyses to clarify this relationship.
  2. Consider reducing the number of figures derived from bioinformatic analyses. Figures that do not directly support the main conclusions may distract the reader and dilute the impact of the manuscript.
  3. Please ensure color consistency across all figures—especially when distinguishing IGF2BP3-high and -low expression groups.

Author Response

Comments1: One major concern is the claim that IGF2BP3 serves as a prognostic marker would be more convincing with additional validation. It would be helpful to compare AUCs from models using only demographic/clinical information (e.g., gender, age, stage, T, N) versus models that also include IGF2BP3 expression.

Response1: We are very grateful for your valuable comments.  By comparing the predictive efficacy (AUC) of the only demographic/clinical information characterization model with the inclusion of the IGF2BP3 expression model, it is possible to more directly demonstrate the value of the additional contribution of IGF2BP3 to outcome prediction, which is essential to establish its validity as an independent prognostic marker. In the current ROC curve analysis (Supplementary Figure 1F), we have actually included part of the relevant information, which provides only demographic/clinical information for the comparison the predictive ability (AUC) of the characteristic variables were: age (AUC=0.518), gender (AUC=0.454), clinical stage ( stage, AUC=0.687), T-stage (AUC=0.648), and N-stage (AUC=0.631). More importantly, univariate analysis showed that the AUC of IGF2BP3 expression by itself (0.634) was already significantly higher than that of sex (0.454) and age (0.518), and was comparable to or close to the key clinical staging metrics of T (0.648), N (0.631), and overall stage (0.687). This suggests that the predictive validity of IGF2BP3 alone rivals or even surpasses some important clinical parameters. In addition, the predictive efficacy of our developed composite risk score model (risk score) integrating multiple variables including IGF2BP3 and clinical information achieved the highest AUC value of 0.717. This is significantly better than any single variable presented in the figure (highest univariate value of the stage was 0.687), providing strong evidence that using the IGF2BP3 information in conjunction with other variables, the synergistically did improve the predictive accuracy of the overall model. We have added its contents to lines 267-271 in the manuscript and labeled them in red.

Comments2: The scRNA-seq data presented in the Fig1 is difficult to interpret and lacks clear connection to IGF2BP3 expression. The authors should strengthen the biological rationale or provide more focused analyses to clarify this relationship.

Response2: We sincerely appreciate your insightful comments regarding the need to clarify the relationship between epithelial cells and IGF2BP3 expression in our single-cell data. In the updated tsne diagram (Fig 1H), we clearly circled the epithelial cell populations with red boxes to help readers quickly identify the target cell populations and avoid ambiguity. In Figure 1I, we would like to see the distribution of IGF2BP3 expression among various cell subpopulations, combined with Fig 1J IGF2BP3 expression in epithelial cells Normal and Tumor, tumor epithelial cells have significantly higher IGF2BP3 expression than that in the normal group (Fig 1J), reinforcing the biological basis for the subsequent functional studies.

Comments3: Consider reducing the number of figures derived from bioinformatic analyses. Figures that do not directly support the main conclusions may distract the reader and dilute the impact of the manuscript.

Response3: Thank you very much for your valuable comments. We agree with you and highly value your suggestion to streamline the graphs and charts related to bioinformatics analysis to avoid irrelevant information distracting the reader and weakening the impact of the core conclusions of the article. After careful consideration, we understand that the experimental data contained in these graphs are complementary to support the study conclusions, and direct deletion may lose some of the valuable information. Therefore, we chose to optimize the graphs: in Figure 6C and Figure 6D, red borders were added to the results section for CD4 Tem and Th2 cells. We believe that this treatment maintains the integrity of the data, enhances the pointing of the graphs to the core findings, minimizes the distraction of non-core information, and echoes your suggestions for enhancing the impact of the manuscript.

Comments4: Please ensure color consistency across all figures—especially when distinguishing IGF2BP3-high and -low expression groups.

Response4: Thank you for pointing out the color consistency issue. We have aligned the colors of the IGF2BP3-high and IGF2BP3-low expression groups to red (high) and blue (low) as per your suggestion, and have ensured that the following diagrams strictly follow this standard: Figure 2H, Figure 2E, Figure 6C, and Figure 6D (the original diagrams have been corrected).

Round 2

Reviewer 1 Report

Comments and Suggestions for Authors

The authors have addressed my concerns; therefore, I endorse the publication.

Reviewer 2 Report

Comments and Suggestions for Authors

Accept in present form

Reviewer 3 Report

Comments and Suggestions for Authors

Thank you for the response. The authors' response did not adequately address all my concerns. The irrelative figures still account for the majority of this manuscript. I prefer to reject it.